# Brain tumour segmentation in fused MRI-PET images with permutate U-Net framework

Yepuganti Karuna[1], Venu Allapakam[2,3], S. Priyanka[3], SK. Riyaz Hussian[4], Peet Nalwaya[3], Saladi Saritha[1]*

1 School of Electronics Engineering, VIT-AP University, Amaravati, Andhra Pradesh, India, 2 Siddarth institute of Engineering and Technology, Puttur, Tirupathi, Andhra Pradesh, India, 3 School of Electronics Engineering, Vellore Institute of Technology, Vellore, Tamil Nadu, India, 4 Department of Electronics and Communication Engineering, RGUKT, Nuzvid, Andhra Pradesh, India

* saritha.saladi@vitap.ac.in

## Abstract

Brain tumor segmentation from MRI's and PET has always been a challenging and time-consuming phase for radiologists, due to low sensitivity boundary region pixels in this image modality. Deep learning-based image segmentation is the hot research topic in recent days. Among all other deep learning models, U-Net-based variants are the most used models to segment medical images with respect to different modalities. In this paper, a Permutate version of the U-Net architecture was designed that precisely and automatically detects the boundaries of the tumour area and segments tumour regions from the fused image. There are two stages to the proposed work. In the first stage Principal component analysis (PCA) is used to fuse the MRI-PET images to enhance the fused image's quality and improved interpretation. Later, a Permutate U-Net architecture is employed to precisely segment tumour region from the fused image. Further designed model performance is assessed using Dice Coefficient, intersection over union score (IoU) and accuracy with brain tumour segmentation challenge BraTS datasets of 2015, 2020 and 2021. Our proposed method demonstrates promising results that are superior to existing deep learning model and comparatively higher than the existing methods.

## 1. Introduction

The brain is sophisticated bodily organ that manages a wide range of functions, including thought, movement, respiration and every bodily regulation process. It also coordinates the body's responses to stimuli, enabling us to move, communicate, and interact with the world around us; thus, brain's health plays a very crucial role in the proper regulations, functioning of overall body. In today's world, it has been a noticeable change in habits and lifestyles, leading to health problems, including tumours treated as threats to mankind. Cancer was one of the foremost causes of mortality

**Data availability statement:** The data is publicly available data here: https://www.kaggle.com/datasets/mateuszbuda/lgg-mri-segmentation, https://www.med.harvard.edu/aanlib/,BraTS 2020 Dataset (Training+Validation)(kaggle.com), BRaTS 2021 Task 1 Dataset (kaggle.com).

**Funding:** The author(s) received no specific funding for this work.

world-wide in 2020, approximately 10million deaths. A WHO report listed brain tumours as one of the most prevalent cancers and their incidence is rising. The International Association of Cancer Registries (IARC) estimates that over 24,000 people lose their lives to brain tumours each year [1–2].

Brain cancer is predicted to develop annual growth rate of 1.11% by 2030, according to a Delve Insight analysis. An estimate is that 40,000–50,000 persons in India receive a brain tumour diagnosis each year. Children make up approximately 20% of these [3]. The number could be far higher, according to experts, as many incidents, particularly in rural areas, go unreported.

The key to treating a brain tumour is spotting it accurately. The variety of distinct types of brain tumours makes diagnosis very difficult. A good classification process can provide suitable care, which in turn depends on the effective segmentation of the tumour. Medical images of different modalities are used for the detection of brain tumours. Different modalities provide unique and specific information, with some offering only structural information while others provide functional data. However, for proper diagnosis and analysis, more detailed information with improved imaging quality is key factor [4]. In most cases, MRI and PET images are the most frequently used, because the processes have no harmful effect on the human body and provide precise information for diagnosis.

MRI images are widely used in the detection of various medical conditions [5]. These images provide highly detailed and precise features of the internal organs, including the brain, without subjecting the patient to radiation injury, which helps in the early detection of the cause and intern helps the doctor plan and execute appropriate treatment. They are particularly helpful in identifying the kind, size, and location of brain tumours as well as other illnesses such as multiple sclerosis, stroke, and joint issues. Overall, MRI images are of great importance in the medical domain.

On the other hand, PET scans are generated by gamma rays that are emitted by a small radioactive material to determine the internal structures. PET images are widely used in the medical domain for the analysis of a variety of medical conditions [6]. PET scans can assess the metabolic activity of tumours and aid in the detection of cancer. PET scans reveal the biochemical activity of tissues and organs, enabling medical professionals to identify cellular abnormalities. For patient treatment planning and management, PET scans are very helpful in identifying tumour recurrence and metastasis.

Brain tumour identification and segmenting the tumour area rely heavily both on MRI and PET images, but due to low sensitive boundaries pixel regions of this image modality it is very hard to accurately segment it. Additionally, manual segmentation process is computationally intensive and time-consuming due to complex tumour shapes, which mainly depends on expertise. Shortage of experience radiologist also challenges issue need to be consider and traditional segmentation methods always results inaccurate segmentation.

Therefore, the contribution of this work is to combine both imaging modalities to generate fused output using appropriate fusion method that results in improved imaging quality in terms of both qualitatively and quantitatively [7–8]. And In addition, this

research will look at various advances and improvements in the U-Net design as well as present techniques, all with the objective of emphasizing U-Net's ongoing potential for improving brain tumor segmentation performance.

The rest of the paper is organized as follows: The literature survey is presented in Section II. The proposed methodology along with a) the Data Set b) the Pre-processing Steps c) the PCA algorithm for fusion and d) the Proposed U-Net model is described in section III. Evaluation metrics are discussed in section IV. Section V deals with the results part. Finally, discussion and conclusion are presented in Section VI

## 2. Literature survey

Brain tumour segmentation is always thought-provoking task due to low sensitivity boundary region pixels in this image modality. Conventional methods of Segmenting the tumour region always results inaccurate results and Traditional U-Net model lags in accurate segmentation. Several works carried out in past based on various machine & deep learning techniques that focus on segmentation of brain tumour. Here, reviews some of the works pertinent to the medical images, followed by the identification of their shortcomings.

In stark contrast to the conventional manual learning method, automatic segmentation based on deep learning (DL) techniques became very popular and CNN frameworks with U-net is most significant one.The benefit of this network structure is that it not only processes medical images successfully and objectively evaluates them but also helps to increase the accuracy of diagnoses. It can accurately segment the required features. The U-net network structure was used in this article to address a few issues with picture shadow and overlap for six different imaging systems. In addition to being a DL innovation has laid the groundwork for accurate pathological diagnosis by doctors [9].

Norouzi, Alireza, et.al explain the various modern image processing technique which are frequently employed in medical image analysis. They discuss four different segmentation techniques: the region-based approach, clustering, classification, and hybrid methods. For several medical images, it was shown that the fuzzy C-mean has greater accuracy. The hybrid approaches combine the region and boundary attributes. The graph-cut method offers the best N-dimensional segmentation results using cost function [10].

Hesamian, Mohammad Hesam, et.al offer a critical evaluation of popular approaches to medical image segmentation making use of deep learning techniques. They have discussed practical solutions to a range of problems [11].

The U-net design, one of the extensive DL architectures for image segmentation, employed in this study by Skourt, BrahimAit, et.al for lung CT image segmentation. The structural design is comprised of an encoder path that contracts to extract high-level information and a decoder path that expands symmetrically to restore the required information. This network outperforms many others and can be trained with a small number of images. The goal of this work was to segment lung nodules [12].

Dhana Chandra, Nameirakpam, et.al presented a segmentation model, using the unsupervised K-means clustering technique. A median filter is used to enhance the segmented image. They claim that the clustering technique has better segmentation accuracy. As future work, they suggested a morphological operation to enhance the attributes of output image [13]

Huang, Hong, et al. proposed a combined FCM clustering algorithm with rough-set theory. First, segmentation findings of FCM are used to build an attribute value table, and the picture is then divided into numerous small sections based on the indistinguishable relationship between the attributes. The next step is to determine the weight values for each attribute, which are then utilized as the foundation for calculating the differences across regions. Using an equivalent connection, each region's similarity is assessed. According to the experimental findings, they propose better segmentation outcomes while reducing error rates when compared to the FCM method [14].

Chowdhary, et al have described a few image segmentations and feature extraction techniques that are relatively new in medical imaging, especially for mammography. If mammograms are utilized without improvement, the radiologists may make wrong judgments. Principal and independent component analysis are frequently employed for

dimensionality reduction. Clustering is the best segmentation technique out of all of them, and it is employed in a variety of applications [15]

The hybrid approach suggested by Özyurt et.al makes use of neutrosophy and convolutional neural networks. During the classification stage, CNN collected the characteristics of the segmented brain images and categorized those using SVM and KNN classifiers. The results showed that CNN features have excellent classification performance across a range of classifiers. According to experimental findings, CNN features performed better at classification with SVM [16].

A depth variation analysis of the U-NET architecture for brain tumor segmentation was proposed by Jena, Biswajit, et al. in 2023 using the BraTS-2017 and BraTS-2019 datasets, which include High-Grade Glioma (HGG) and Low-Grade Glioma (LGG) MR Scan. An effective examination of the depth variation of the U-NET architecture—that is, after removing various layers—has been carried out in this work. The cross-validation replication cohort had very low dice coefficient of at least 0.8895 and as high as 0.8911, while the discovery cohort had a dice coefficient of at least 0.8866 and as high as 0.8887 [17].

In 2024, Lin, Shu-You, and Chun-Ling Lin presented a U-Net and Efficient Net v2 for brain tumour segmentation using the BraTS-2019 data set. In this work, the traditional U-net architecture is combined with an Efficient Net V2 encoder to enhance the model performance of the U-net architecture. The results of the experiment demonstrated that the model successfully segmented the tumor region with a minimal dice score of −0.0866, a dice similarity coefficient (DSC) of 0.9133, and a high accuracy of 0.9977 [18].

Using the Brats2019 dataset, Zhang, Yajie, et al. (2024) presented an attention mechanism to help the network focus on significant regions, overcoming the limitations of conventional U-net based segmentation techniques in handling very small targets and fuzzier boundaries. The proposed model makes use of multiscale feature fusion strategy to improve the efficacy of network segmentation at various scales. The experimental results demonstrate that the proposed methodology exhibits superior speed and efficiency. But the model results very low dice coefficients for the multiple branch TS-U-net model were 0.876, 0.868, and 0.814 in the tumour subregions of WT, TC, and ET, respectively [19].

Thus, there are various techniques available for detection of brain tumours. However, after going through the above works, a few drawbacks have been identified and mentioned below.

- The laborious process of manually segmenting with huge amount of data and shortage of experienced radiologist, always results in poor delineation.

- Delineation of tumour boundaries with accurate segmentation is very crucial and always a challenging task.

- Traditional U-Net model lags in accurate segmentation of region of interest of brain tumours with very low performance metric.

## 3. Proposed fusion based segmentation model

To addresses the above limitation, The proposed Permutate U-Net framework with additional layer in encoder and decoder are designed to perform segmentation task effectively from fused-MRI-PET images. The overall proposed configuration is categorized into two stages. Firstly, the fusion of MRI-PET images using PCA is taken up, The PCA-based fusion method combines the features of both MRI & PET images, resulting in more comprehensive and informative image with improved imaging quality for brain tumour segmentation.

Secondly, a modified U-Net for accurate segmentation from the resultant fused image is proposed. The block diagram of the proposed framework is indicated in Fig 1. The foremost motive of the work is to develop a custom U-Net model from scratch that can effectively segment tumour area from fused MRI and PET images.

The U-Net model processes the fused image to provide accurate and reliable results in segmenting any potential tumour. The flowchart of the proposed model is present in Fig 2. The dataset, pre-processing steps, image fusion using PCA, and the proposed U-Net model are present in Section 3. A-D respectively

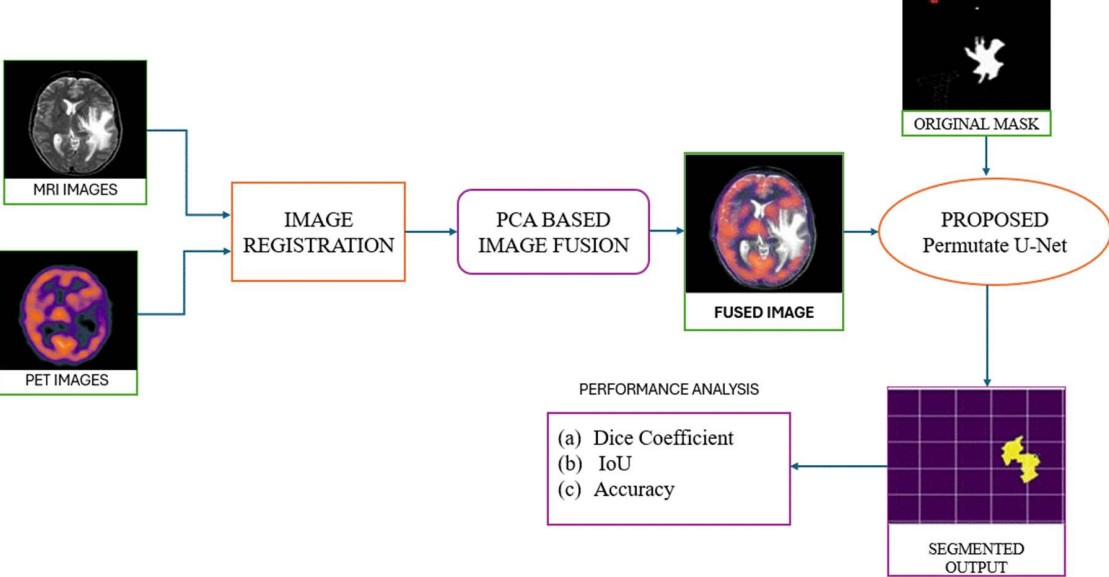

**Fig 1. Block-diagram of the proposed framework.**

First, the two inputs, MRI, and PET images, are taken, and these images are processed through an image registration process that involves aligning the co-ordinates of both the images. Then, these images are used for fusion. The PCA algorithm is used for fusing the MRI & PET images. Then, the fused image is given as an input to the modified U-Net model. This model is trained using the datasets mentioned in Section III.A. The proposed U-Net model uses a 5-level encoder and decoder layer configuration to obtain the segmented output image. Parameters, such as accuracy, dice-coefficient and IoU are used for evaluation purpose.

## 3.1 Dataset

In general, pre-made data sets are preferred when working with medical images. These data sets have been made available to the researchers after the images are labelled by experts. One of these is the BraTS (2015, 2020, and 2021) "Multimodal-Brain-Tumour-Segmentation Challenge" datasets. In the proposed work, to test and evaluate the model performance we consider the above publicly available data set, which has 110 patient scanned MRI and PET brain tumour images. We considered 88 images for training and 22 images for testing, input dimensions 256x256, we chose to make use of only images of tumours for training to make sure the network could find tumour features quickly. Next, we included all images in the network learning standardization.

**Ethical statment.** This study used publicly available or simulated data and did not involve human participants or animals. Therefore, ethical approval and informed consent were not required.

## 3.2 Pre-Processing

Image pre-processing is an important phase when dealing with numerous datasets, since it improves the image quality and assists in identifying the finest features. To increase the eminence of the images, there exist various methods such as scaling, and intensity normalization are employed. To reduce noise, without losing precise information, noise filters are utilized. Below is a thorough explanation on image resizing, normalization, and augmentation which are some related pre-processing steps.

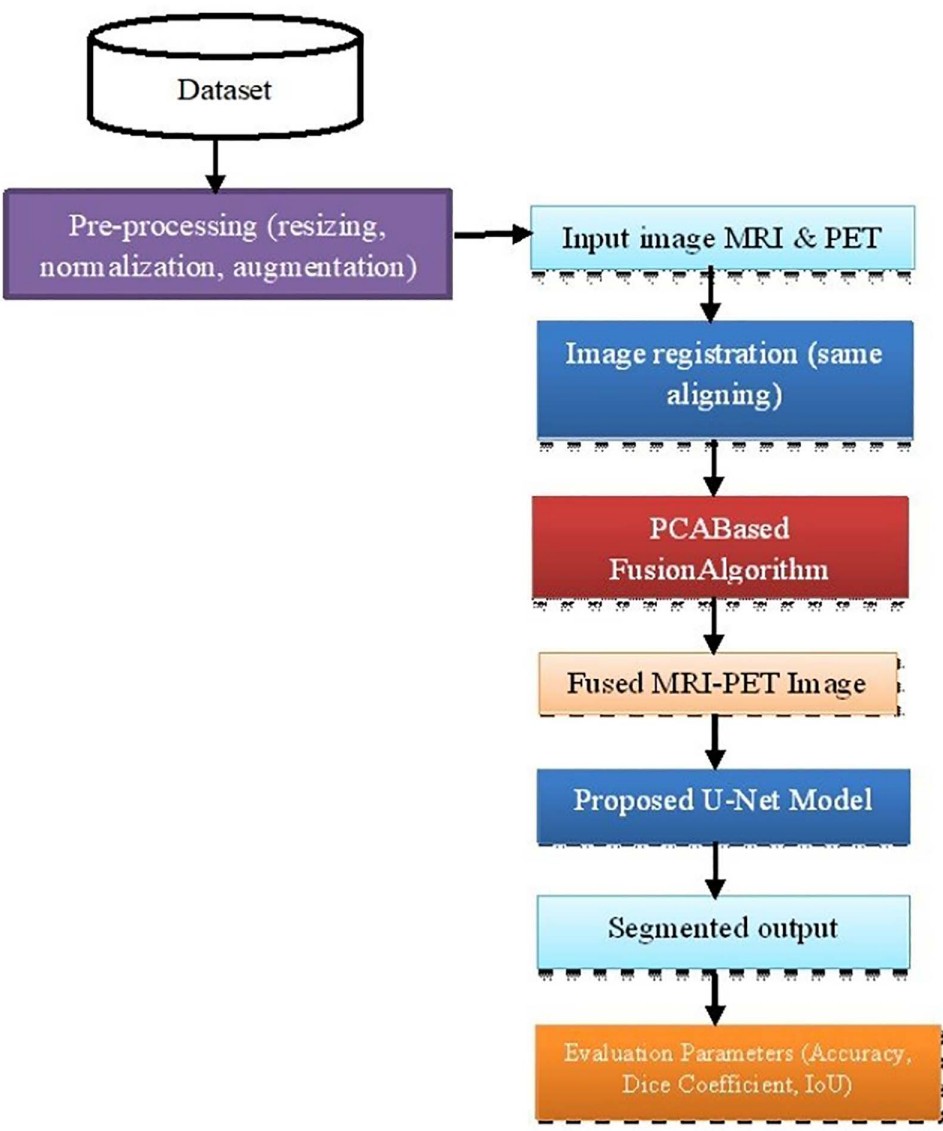

**Fig 2. Flow chart of the proposed algorithm.**

**3.2.1 Image resizing.** Image resizing is a critical and most important pre-processing step in computer vision applications. Resizing is the process of converting all the images to a fixed standard or smaller images that accelerate the model training in machine learning and DL techniques. This process helps in facilitating the processes (comparison and analysis). All the images are resized to standard necessities of 256x256 to make the further process much easier. A 220x220 image size in Fig 3(a) has been resized to the required 256x256 size in Fig 3(b).

**3.2.2 Normalization.** Normalization involves converting all the pixel values of an image into a range [0–1 by dividing them by 256 as we consider 8-bit images. (for 8-bit image it is 256, for 12- and 16-bit image it will be 4095 and 65535 respectively as the maximum value). It is sometimes called histogram stretching or contrast stretching. This adjustment helps to enhance the brightness and contrast, suitable for subsequent processes, as illustrated in Fig 4.

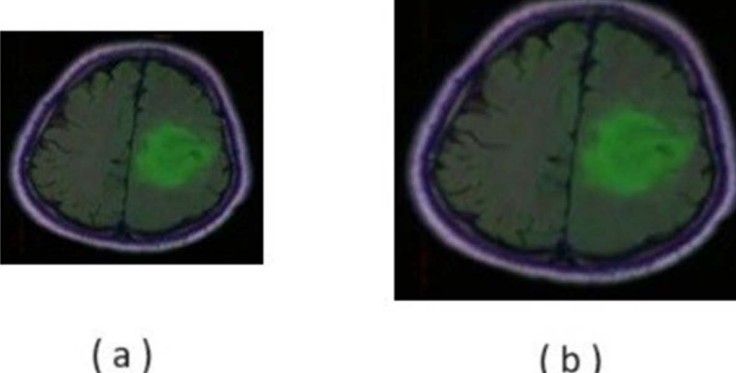

**Fig 3. Image resizing: (a) Input image (b) Resized image.**

**3.2.3 Augmentation.** It is an image pre-processing technique aimed at making the mode l more robust. It is defined as the process of generating or producing a new version of image from the given image to increase its diversity. It is mainly used for expansion of data sets artificially through techniques like rotation, brightness adjustment, zooming in/out, and various flipping and shifting in horizontal and vertical directions. It is also employed on an image to produce new variations of the same image, thus increasing the dataset's diversity, and strengthening the model, as shown in Fig 5. After augmentation the size of the data set was raised to 110 from 92.

### 3.3 MRI-PET image fusion using PCA

In this section we present the PCA algorithm for dimensionality reduction followed by fusion of MRI & PET images.

Step-1: Let the images be $[I_n] \in R^{W*H}$ for n = 1,2…. N. Form a vector $\mathbf{x}_n$ = vector $(I_n) \in R^D$, where D = W*H is the dimension of the image. Form a matrix $[\mathbf{X}]= [\mathbf{x}_1, \mathbf{x}_2,..\mathbf{x}_N] \in R^{D*N}$.

Step-2: For each row of the matrix **X**, compute mean and standard deviation. From every element in row, subtract the mean then divide by standard deviation such that a normalized matrix $[\widetilde{X}]$ is formed.

Step-3: Compute the covariance matrix $S = \frac{1}{N}\widetilde{X}\widetilde{X}^T \in R^{D*D}$. Perform Eigen Value Decomposition (EVD) of **S** such that S = U∑U$^T$. Columns of **U** – Eigenvectors and ∑ is diagonal matrix with Eigen-values as the diagonal elements.

Step-4 Select the first K Eigen vectors of **U** corresponding to the K maximum Eigen values. Form a matrix.

$\mathbf{B}$= $[\mathbf{u}_1, \mathbf{u}_2, …. \mathbf{u}_K] \in R^{D*K}$. Compute $\hat{X} = BB^T\widetilde{X}$, where $B^T\widetilde{X} \in R^{K*N}$ and $\hat{X} \in R^{D*N}$. The elements of each row of $\hat{X}$ are multiplied by the corresponding standard deviation and mean is added to restore the dynamic range of rows of $\hat{X}$.

Step-5: Each column of $\hat{X}$ is converted back to image format.

The above steps are repeated for both MRI & PET images. Finally, pixel wise fusion of reduced dimension images can be taken up. These fusion algorithmic steps are shown in Fig 6. The PSNR and SSIM values for the fused images using the PCA algorithm are found to be comparatively better than those of the other fusion algorithms. The qualitative and quantitative results of different fusion algorithms are shown in fig and Table-3 respectively.

### 3.4 Permutate U-Net framework for brain tumour segmentation

The Permutate U-Net architecture originated from traditionally available CNN network and is specially developed for medical images. A general CNN network takes an image as input and classifies it with a single label. However, in cases of medical images, it is not enough to classify whether there is an abnormality or not; it is also necessary to localize the

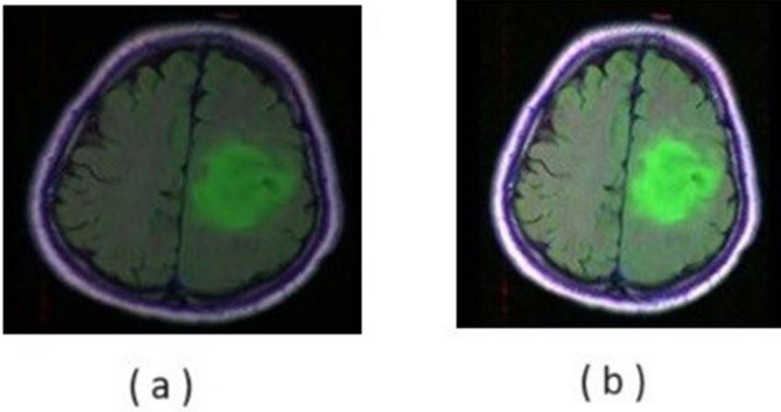

**Fig 4. Normalization of data (a) input image (b) image after normalization.**

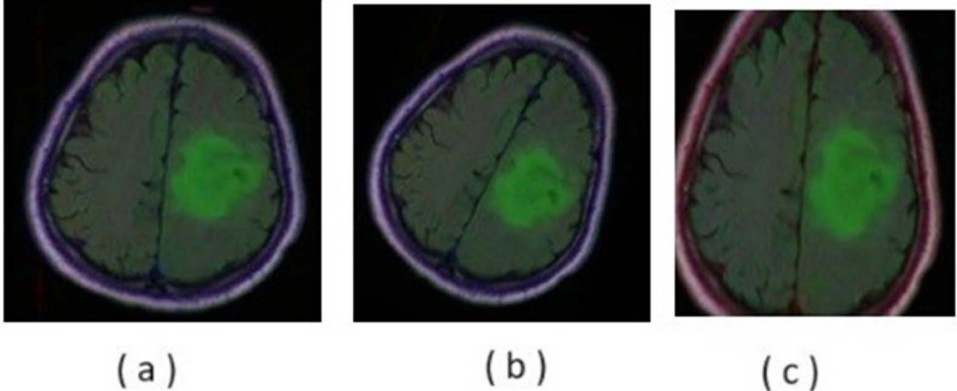

**Fig 5. Augmentation (a) input image (b) magnified image (c) image after horizontal rotation.**

region of the abnormality. This is where U-Net has the advantage over CNN. U-Net architecture performs classification on every pixel of the image, allowing it to localize and differentiate abnormal borders.

This paper proposes the use of a Permutate U-Net model to segment brain tumors automatically as shown in Fig 7. The Permutate U-Net model has been custom-designed with layers, which were specifically chosen through several experiments to achieve the desired results. The proposed U-Net model features 5 down sampling layers, 5 up sampling layers, and a middle layer, increasing the number of skip connections and filters. The down sampling layers consist of a combination of convolution, max pooling, and batch normalization, with activation performed after the convolution layer and batch normalization. Similarly, the up-sampling layers contain a combination of transpose convolution, concatenation, and activation and normalization after every concatenation. The use of activation functions and batch normalization play a significant role in improving the performance and steadiness of the model. This helps the model to learn relationships in a better way.

In this way, the model learns the parameters and over fitting is reduced. The model makes use of the ReLU and sigmoid activation functions due to the computational efficiency that they offer. The ReLU activation function speeds up the

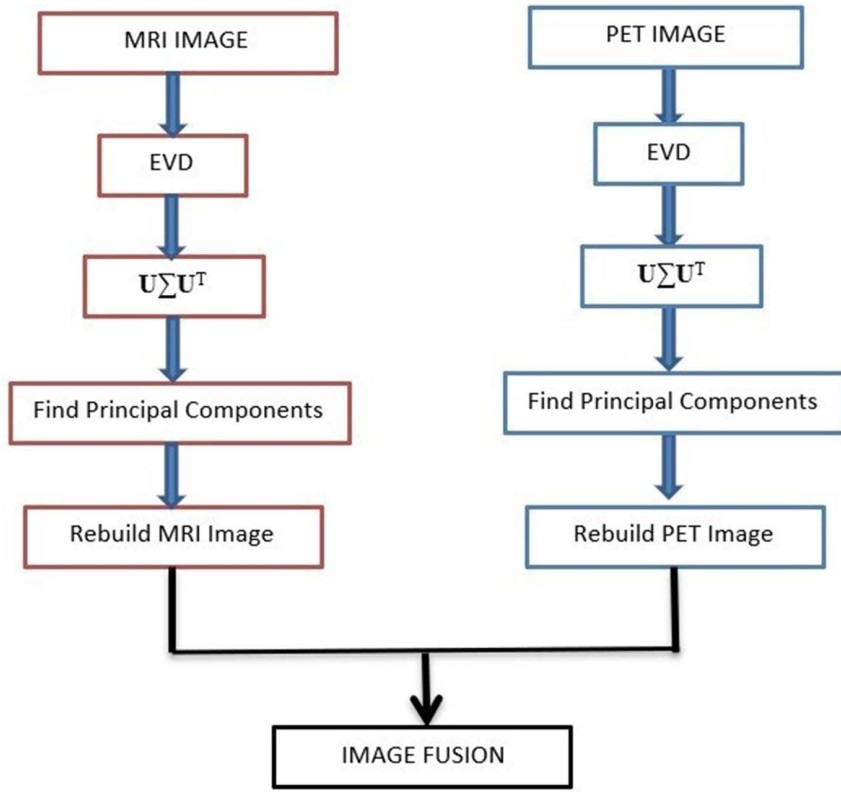

**Fig 6. PCA based MRI-PET image fusion.**

training by providing faster convergence and helps with non-linear mapping. Sigmoid is another activation function that is used.

The dimensions of the image are 256X256X3, and all the layers and hyper parameters are designed according to this input size to achieve the best possible results. The suggested U-Net model offers an effective and efficient method for accurately identifying and isolating tumors, representing a significant development in automatic brain tumor detection. The input image of size 256X256X3 passes through a series of five down sampling blocks, where the image undergoes a combination of two convolution layers followed by activation after every convolution and batch-normalization and a max-pooling layer.

A 3x3 filter, used for convolution, and 2x2 max pooling is performed with a stride of 1. The number of filters is doubled, starting from 64, 128, 256, and so on up to 1024. The filters provide depth to the model.

After passing through the five down sampling blocks, there is a middle layer with 2048 filters and a similar combination of two convolution layers, activation, and batch-normalization layers. The image is passed through this block (8x8x2048). Then, the image is sent through a combination of up sampling blocks. Five up sampling blocks are used in the model. These blocks have a combination of a transposed convolution layer, a concatenation layer, and two convolution layers. The size of the filters used in each block of decoder is reduced from 1024 to 64 in step 2. In the last up sampling block, convolution is performed three times, the activation function used is sigmoid, and the filter count is 1. The input from the middle leg to the decoder leg is 8x8x2048, and the output is 256x256x1. The review of the proposed model is accessible in Table 1.

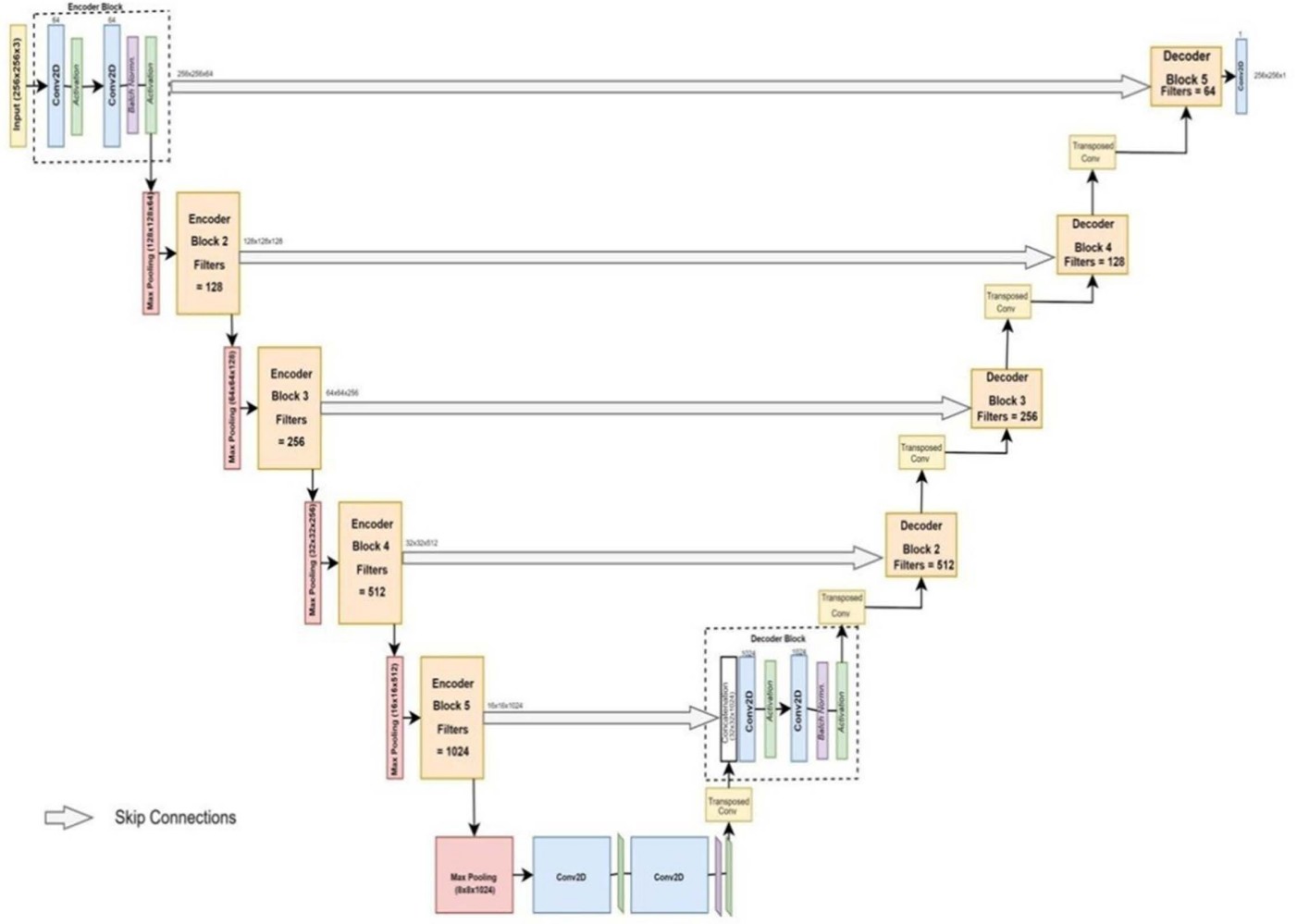

**Fig 7. Proposed U-Net block diagram.**

## 4. Performance evaluation

The proposed model is evaluated with the following metrics, both fusion and segmentation. Fusion performance is analyzed using PSNR & SSIM [20]. These results are presented in Table 2 for different fusion techniques, PCA method of fusion is showing better results both for PSNR and SSIM. To assess the segmentation performance, we evaluate the dice-coefficient, IoU and the accuracy. Table 3. shows comparative results of segmentation parameters.

### 4.1 Peak-Signal-To-Noise-Ratio (PSNR)

For the qualitative analysis and evaluation of the fused image obtained from various algorithms, PSNR and SSIM are taken into consideration. PSNR is an image quality metric, defined as,

$$PSNR = 10\log_{10}\frac{Max_i2}{MSE}$$

(6)

**Table 1. The layers in Proposed U-Net (Total number of trainable parameters are: 122,480,513).**

| Block Name | Laye rName | Input Shape | Filters | Output Shape | Number of Parameters |
|---|---|---|---|---|---|
| Down Sampling Block-1 | Inputimage | 256x256x3 | – | 256x256x3 | 0 |
| | Conv2D | 256x256x3 | 64 | 256x256x64 | 1792 |
| | Activation | 256x256x64 | – | 256x256x64 | 0 |
| | Conv2D | 256x256x64 | 64 | 256x256x64 | 36928 |
| | Batch Normalization | 256x256x64 | – | 256x256x64 | 256 |
| | Activation | 256x256x64 | – | 256x256x64 | 0 |
| Down Sampling Block-2 | Maxpooling | 256x256x64 | 64 | 128x128x64 | 0 |
| | Conv2D | 128x128x64 | 128 | 128x128x128 | 73856 |
| | Activation | 128x128x128 | – | 128x128x128 | 0 |
| | Conv2D | 128x128x128 | 128 | 128x128x128 | 147584 |
| | Batch Normalization | 128x128x128 | – | 128x128x128 | 512 |
| | Activation | 128x128x128 | – | 128x128x128 | 0 |
| Down Sampling Block-3 | Maxpooling | 128x128x128 | 128 | 64x64x128 | 0 |
| | Conv2D | 64x64x128 | 256 | 64x64x256 | 295168 |
| | Activation | 64x64x256 | – | 64x64x256 | 0 |
| | Conv2D | 64x64x256 | 256 | 64x64x256 | 590080 |
| | Batch Normalization | 64x64x256 | – | 64x64x256 | 1024 |
| | Activation | 64x64x256 | – | 64x64x256 | 0 |
| Down Sampling Block-4 | Maxpooling | 64x64x256 | 256 | 32x32x256 | 0 |
| | Conv2D | 32x32x256 | 512 | 32x32x512 | 1180160 |
| | Activation | 32x32x512 | – | 32x32x512 | 0 |
| | Conv2D | 32x32x512 | 512 | 32x32x512 | 2359808 |
| | Batch Normalization | 32x32x512 | – | 32x32x512 | 2048 |
| | Activation | 32x32x512 | – | 32x32x512 | 0 |
| Down Sampling Block-5 | Maxpooling | 32x32x512 | 512 | 16x16x512 | 0 |
| | Conv2D | 16x16x512 | 1024 | 16x16x1024 | 4719616 |
| | Activation | 16x16x1024 | – | 16x16x1024 | 0 |
| | Conv2D | 16x16x1024 | 1024 | 16x16x1024 | 9438208 |
| | Batch Normalization | 16x16x1024 | – | 16x16x1024 | 4096 |
| | Activation | 16x16x1024 | – | 16x16x1024 | 0 |
| MiddleLayer | Maxpooling | 16x16x1024 | 1024 | 8x8x1024 | 0 |
| | Conv2D | 8x8x1024 | 2048 | 8x8x2048 | 18876416 |
| | Activation | 8x8x2048 | – | 8x8x2048 | 0 |
| | Conv2D | 8x8x2048 | 2048 | 8x8x2048 | 37750784 |
| | Batch Normalization | 8x8x2048 | – | 8x8x2048 | 8192 |
| | Activation | 8x8x2048 | – | 8x8x2048 | 0 |
| Up Sampling | Conv2D | 8x8x2048 | 2048 | 16x16x1024 | 8389632 |

*(Continued)*

**Table 1.** (Continued)

| Block Name | Laye rName | Input Shape | Filters | Output Shape | Number of Parameters |
|---|---|---|---|---|---|
| Block-1 | Transpose | | | | |
| | Concatenate | 16x16x1024 | – | 16x16x2048 | 0 |
| | Conv2D | 16x16x2048 | 1024 | 16x16x1024 | 18875392 |
| | Activation | 16x16x1024 | – | 16x16x1024 | 0 |
| | Con2D | 16x16x1024 | 1024 | 16x16x1024 | 9438208 |
| | Batch Normalization | 16x16x1024 | – | 16x16x1024 | 4096 |
| | activation | 16x16x1024 | – | 16x16x1024 | 0 |
| Up Sampling Block-2 | Conv2D Transpose | 16x16x1024 | 1024 | 32x32x512 | 2097664 |
| | Concatenate | 32x32x512 | – | 32x32x1024 | 0 |
| | Conv2D | 32x32x1024 | 512 | 32x32x512 | 4719104 |
| | Activation | 32x32x512 | – | 32x32x512 | 0 |
| | Con2D | 32x32x512 | 512 | 32x32x512 | 1179904 |
| | Batch Normalization | 32x32x512 | – | 32x32x512 | 1024 |
| | activation | 32x32x512 | – | 32x32x512 | 0 |
| Up Sampling Block-3 | Conv2D Transpose | 32x32x512 | 512 | 64x64x256 | 262400 |
| | Concatenate | 64x64x256 | – | 64x64x512 | 0 |
| | Conv2D | 64x64x512 | 256 | 64x64x256 | 1179904 |
| | Activation | 64x64x256 | – | 64x64x256 | 0 |
| | Con2D | 64x64x256 | 256 | 64x64x256 | 295040 |
| | Batch Normalization | 64x64x256 | – | 64x64x256 | 512 |
| | activation | 64x64x256 | – | 64x64x256 | 0 |
| Up Sampling Block-4 | Conv2D Transpose | 64x64x256 | 256 | 128x128x128 | 65664 |
| | Concatenate | 128x128x128 | – | 128x128x256 | 0 |
| | Conv2D | 128x128x256 | 128 | 128x128x128 | 295040 |
| | Activation | 128x128x128 | – | 128x128x128 | 0 |
| | Con2D | 128x128x128 | 128 | 128x128x128 | 73792 |
| | BatchNormalization | 128x128x128 | – | 128x128x128 | 256 |
| | activation | 128x128x128 | – | 128x128x128 | 0 |
| Up Sampling Block-5 | Conv2D Transpose | 128x128x128 | 128 | 256x256x64 | 16448 |
| | Concatenate | 256x256x64 | – | 256x256x128 | 0 |
| | Conv2D | 256x256x128 | 64 | 256x256x64 | 73792 |
| | Activation | 256x256x64 | – | 256x256x64 | 0 |
| | Con2D | 256x256x64 | 64 | 256x256x64 | 36928 |
| | Batch Normalization | 256x256x64 | – | 256x256x64 | 256 |
| | activation | 256x256x64 | – | 256x256x64 | 0 |
| | Conv2D | 256x256x64 | 1 | 256x256x1 | 65 |

**Table 2. Comparative-Analysis of proposed fusion model with other fusion techniques using PSNR and SSIM.**

| Fusion Algorithms | With MRI Image | | With PET Image | |
|---|---|---|---|---|
| | PSNR | SSIM | PSNR | SSIM |
| Discrete Wavelet Transform | 32.711 | 0.792 | 31.087 | 0.753 |
| Guided Filter Algorithm | 31.078 | 0.671 | 31.080 | 0.742 |
| Averaging Technique | 31.479 | 0.785 | 31.358 | 0.767 |
| Fast Filtering | 31.493 | 0.741 | 31.442 | 0.681 |
| **Principal Component Analysis** | **35.312** | **0.851** | **33.220** | **0.827** |

**Table 3. Comparative analysis with existing works.**

| Author & Year | Methodology | DC | IoU | ACC |
|---|---|---|---|---|
| Baid, U.andS.Talbar.& 2016 [21] | Fuzzy C– mean clustering | 0.750 | 0.66 | – |
| Kamnitsas, Konstantinos, et al & 2017 [22] | 3D CNN with fully connected CRF | 0.85 | – | 0.88 |
| Selvapandian, A and K.Manivannan& 2018 [23] | Morphological functions | – | – | 0.992 |
| Zhao, Xiaomei, et al. & 2018 [24] | Integrated FCNNs and CRFs | 0.84 | – | 0.820 |
| Alkassar, et al & 2019 [25] | CNN Model | 0.774 | 0.80 | 0.977 |
| Teki, Satyanarayana Murthy, et al & 2019 [26] | U-net Based Adversarial Networks | – | 0.89 | 0.940 |
| Gadosey, Pius Kwao, et al & 2020 [27] | Stripped-Down U-Net (SD-U-Net) | 0.827 | – | 0.986 |
| Preethi, et al. & 2021 [28] | SMO-DNN (weighted K-means algorithm) | 0.853 | 0.75 | 0.930 |
| Al-Dabagh,et al. & 2021 [29] | C-mean clustering and seeded region method | – | 0.82 | 0.900 |
| Haq, Imran Ul, et al & 2022 [30] | BTS-GAN | 0.85 | 0.77 | – |
| Guvenc et al.'s & 2023 [31] | Traditional U-Net architecture | 0.739 | 0.59 | 0.982 |
| Proposed Method | Permutate U-Net | **0.91** | **0.88** | **0.996** |

## 4.2 Structural-Similarity-Index-MeasureMENT (SSIM)

The other parameter used is SSIM. It stands for structural similarity index measures. It is a measure of how close the output image is to the input image. SSIM is a measure to quantify the similarity between two images. The SSIM index is a value [-1, 1]. Mathematically,

$$SSIM = \frac{(2\mu_a\mu_b + 1)(2\sigma_{ab} + Z_2)}{(\mu_a^2 + \mu_b^2 + Z_1)(\sigma_a^2 + \sigma_b^2 + Z_2)}$$

(7)

Whereas

• a, b – two images being compared.

• $\mu_a$ and $\mu_b$ – mean, $\sigma_a^2$ and $\sigma_b^2$ – variances, $_{ab}$ is the covariance of a and b respectively.

• $Z_{1\ and}$ $Z_2$- constants.

It is the similarity of the processed image with the input image.

## 4.3 Dice Coefficient (DC)

It is the measure of the check the overlapped regions between the result obtained from segmented image and ground-truth image, with a value range from 0 to 1, where a value of 0 – no overlap, and 1 -perfect overlap

$$DC = 2\,\frac{|I_1 \cap I_2|}{|I_1 + I_2|}$$

(8)

## 4.4 Intersection Over Union (IoU)

The IoU is also known as the Jaccard index. It is used to evaluate the overlap between the predicted and ground truth image. Its value lies between 0 & 1. A value 0 – no similarity and value 1 – perfect similarity.

$$IoU = \frac{|I_1 \cap I_2|}{|I_1 \cup I_2|}$$

(9)

## 4.5 Pixel Accuracy (ACC)

Accuracy is another parameter that is used for model evaluation. It tells how well the model performs. It's the ratio of correct predictions to the overall number of predictions

## 5. Results analysis

The model was executed on Kaggle platform using Python 3, with a GPU utilized as an accelerator for the program. The system has 8GB RAM and a 256GB SSD and Corei5 processor 10$^{th}$ generations. Here we present the simulated outputs of both fused and segmented results.

## 5.1 Fusion results

Fig 8 shows the qualitative comparison of resultant fused MRI-PET Images, using various fusion techniques. Corresponding quantitative results are exhibited in Table 2.

 The graphical representations of the above values for PSNR and SSIM of MRI & PET images are respectively presented in Fig 9. From the comparative analysis graph of various fusion algorithms, it is observed that the PCA based fusion has both better PSNR and SSIM values as compared to other fusion methods. PCA is showing PSNR values of 35.312 & 33.220 and SSIM values of 0.851 & 0.827 with respect to input MRI and PET images respectively. Table 2 shows the Qualitative Comparative Analysis of proposed fusion model with other fusion techniques that were presented. From the above table one can observe the various fused output that are obtained using different fusion algorithms. Among the resultant fused images from various fusion methods, the PCA based fused image provides high image quality. The improved image quality helps with good interpretation and helps in analyse the image at multiple resolutions. The resultant PCA based fused image has enhanced coarse and fine structures helps in effective detection and segmentation of tumours regions with improved performance metrics.

## 5.2 Segmentation results

The proposed U-Net model's segmentation results are reported both quantitatively and qualitatively. Visual analysis of the output segmented image is obtained after training and testing. Fig 10 shows the segmented image output when inputs are taken from the dataset randomly. Fig 11 shows the output results when the custom input is given to the model.

## 5.3 Analysis of parameters evaluation and segmentation results

The model was trained using 100 epochs. Since there had been no discernible improvement in Dice Coefficient, IoU, and Accuracy after 100 epochs, training was stopped at 100 epochs. From the results of training & validation, it can be observed that the model is not overfitting. We have implemented 5-fold cross-validation on the training data (from the

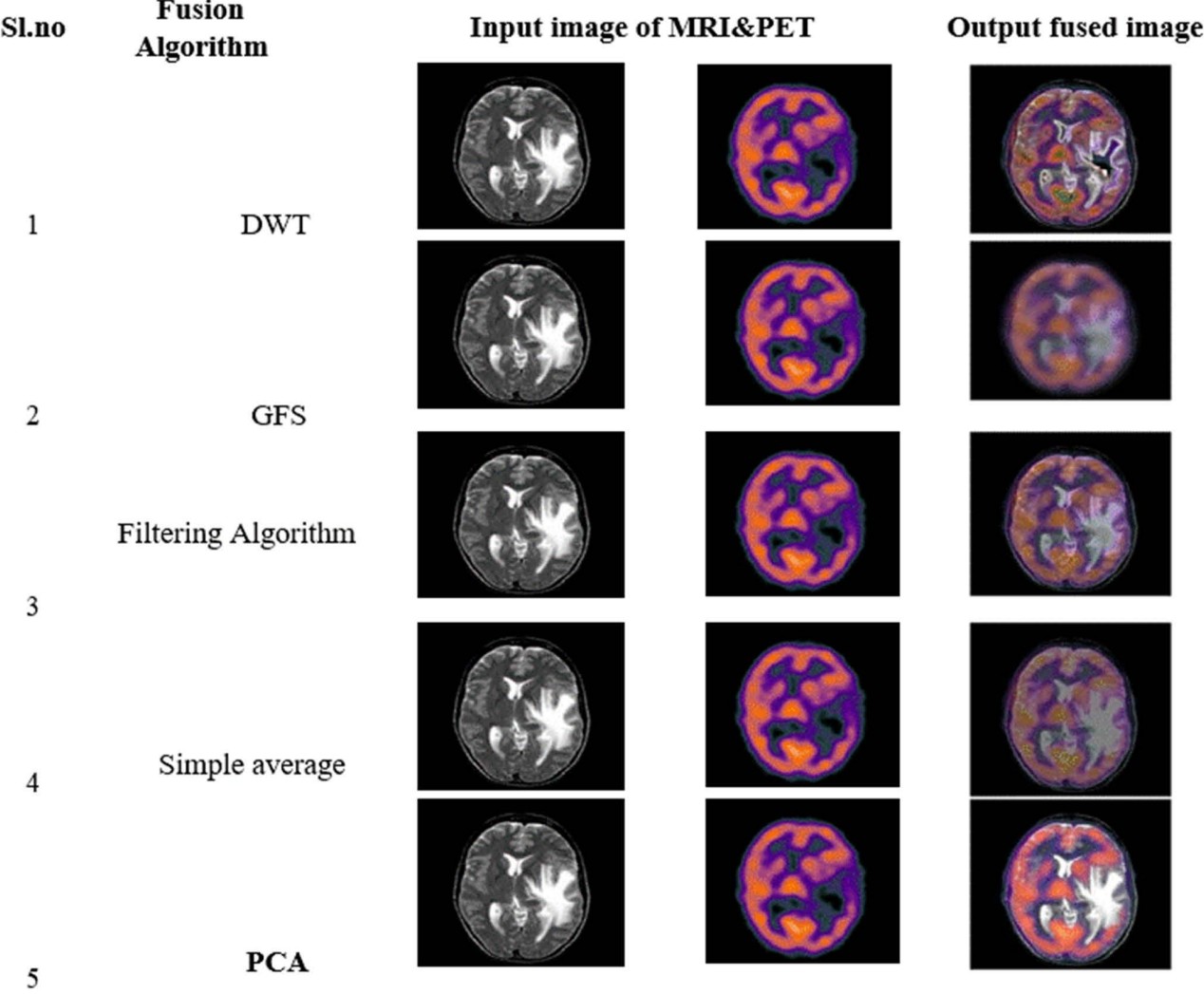

**Fig 8. Qualitative comparison of resultant fused MRI&PET images for various fusion algorithm.**

BraTS dataset) to reduce the likelihood of overfitting and to ensure more reliable performance estimation. The averaged performance metrics like Dice-coefficient, IoU and accuracy across folds are reported in (Tables 5, 6, 7). Fig 12.a, b and c show graphical analysis of training and validation outcomes for the parameter:

### 5.4 Comparative analysis with existing works

The performance analysis and obtained results are presented in Table 3. The brain tumour segmentation model employing fuzzy C-mean cluster [21] over MRI BRATS-2012 Data sets and obtained values of 0.750 and 0.66 for DC and IoU, respectively. An effective multi-scale 3D-CNN model [22] yields dice-coefficients of 0.85- and 0.88-pixel accuracy. A model for segmenting tumour regions in fused MRI-PET images based on morphological functions [23], with model segmentation accuracy of 0.992. Deep learning model incorporating FCNNs and CRFs for brain tumour segmentation [24] the model results moderate dc-0.84 and acc-0.82. CNN model [25] that employs brain image segmentation for MRI images and the model yields DC −0.774, IoU-0.80, and ACC-0.977.

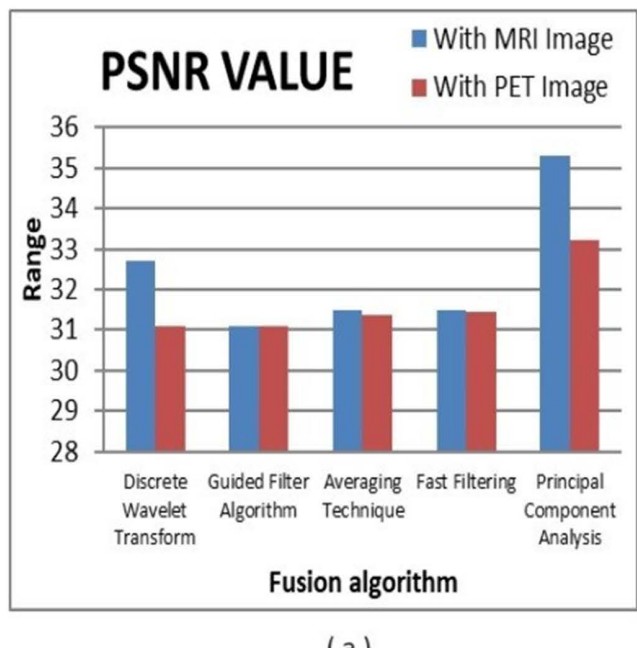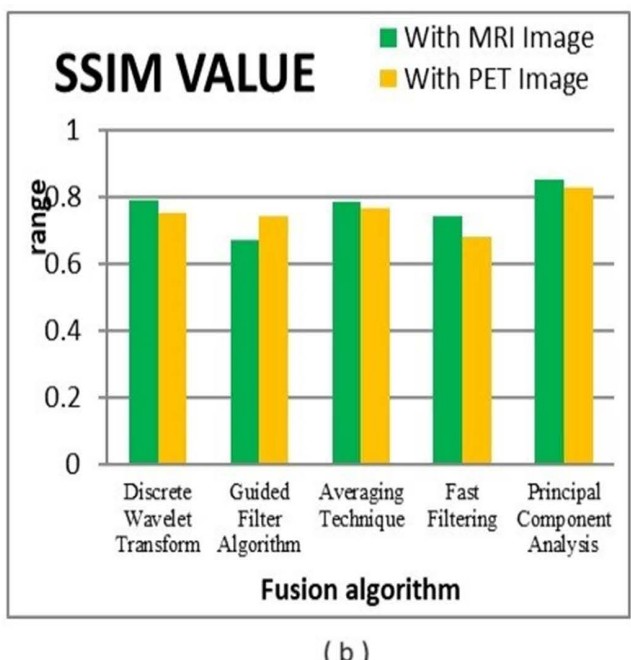

**Fig 9. Graphical representation of various fusion algorithms.** (a) PSNR w.r.t input images, (b) SSIM w.r.t input images.

A novel U-net Based Adversarial Networks model [26], the model yielding a mean accuracy of 0.94 and a high mean intersection over union of 0.89. For brain MRI image segmentation with minimal computing resources, SD-UNET: Stripping down U-net [27]. The model yields extremely high segmentation ACC-0.986 and moderate DC-0.827. An efficient wavelet-based image fusion using PET and MRI images [28], utilizing the SMO-DNN (weighted K-means algorithm). They obtained DC, IoU, and ACC values of 0.853, 0.75, and 0.930, respectively. A technique for segmenting brain MR images utilizing seeded regions and C-mean clustering has been reported by Al-Dabagh et al. [29] with an accuracy of 0.90 and an IoU of 0.82.

A precise and automated method of image segmentation with BTS-GAN utilizing conditional GAN (cGAN) in MRI scans. An average of 0.77 and 0.85 were obtained by proposed model in terms of IoU and Dice scores [30]. The segmentation of brain tumours employing the traditional U-Net architecture [31] for the BRATS −2018 MRI images yielded good accuracy, moderate DC, and low IoU with 0.992, 0.739, and 0.59 respectively. The proposed deep learning model attained DC-0.91, IoU-0.88 and ACC of 0.996, which are significantly greater than the previous result, which are presented in Table 3

### 5.6 Performance comparison of various U-Net with permutate U-Net model

The model's performance was assessed with a variety of publicly accessible brain tumor segmentation challenge BraTS datasets from 2015, 2020, and 2021. Tables 4–6 present the findings of a comparison between the proposed model's performance and those of existing deep learning architecture, for a variety of data sets. For various kinds of BraTS datasets, it is noticed that the proposed U-Net model is superior over existing models in terms of DC, IoU, and ACC performance metrics as shown in graphical representation of Figs 13–15– for various datasets.

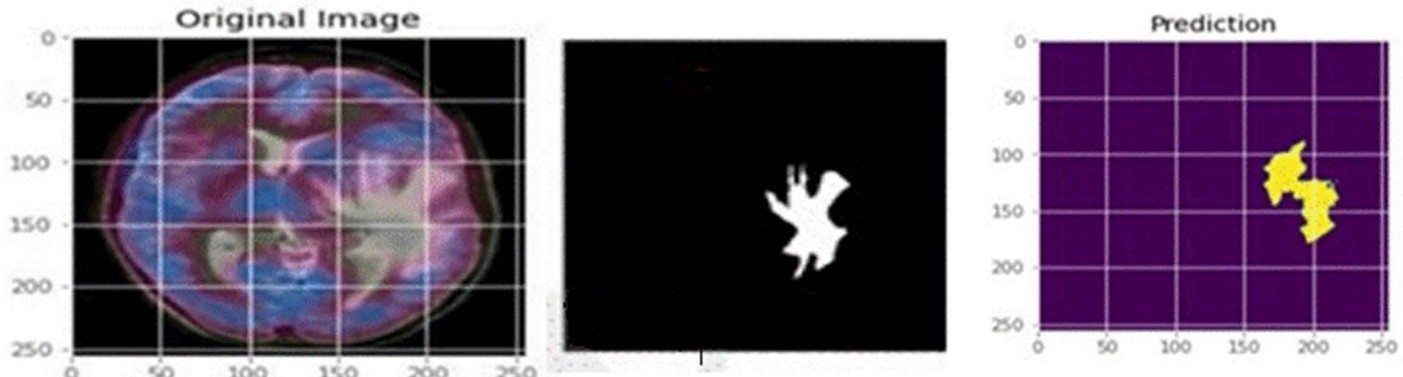

**Fig 10. Segmented tumours regions for randomly selected input images.**

**Fig 11. Segmented tumour regions for custom input images.**

**Table 4. Comparison of the proposed model's outcomes with the existing architecture for the BraTS 2015 datasets.**

| Author details | Methodology | DC | IoU | ACC |
|---|---|---|---|---|
| Alkassar, et al. [25] | CNN | 0.76 | 0.65 | 0.881 |
| Gadosey, Pius Kwao, et al [27] | SD-U-Net | 0.72 | 0.74 | 0.904 |
| Guvenc, et al. [31] | U-Net | 0.80 | 0.78 | 0.938 |
| Raza, Rehan, et al. [32] | Residual U-Net | 0.83 | 0.81 | 0.913 |
| Ahmad, Parvez, et al. [33] | Dense U-Net | 0.81 | 0.79 | 0.926 |
| **Proposed** | Permutate U-Net | **0.89** | **0.86** | **0.987** |

**Table 5. Comparison of the proposed model's outcomes with the existing architecture for the BraTS 2020 datasets.**

| Author details | Methodology | DC | IoU | ACC |
|---|---|---|---|---|
| Alkassar, et al. [25] | CNN | 0.79 | 0.69 | 0.924 |
| Gadosey, Pius Kwao, et al [27] | SD-U-Net | 0.68 | 0.76 | 0.958 |
| Guvenc, et al. [31] | U-Net | 0.84 | 0.82 | 0.946 |
| Raza, Rehan, et al. [32] | Residual U-Net | 0.80 | 0.76 | 0.950 |
| Ahmad, Parvez, et al. [33] | Dense U-Net | 0.87 | 0.82 | 0.933 |
| **Proposed** | Permutate U-Net | **0.93** | **0.89** | **0.991** |

**Table 6. Comparison of the proposed model's outcomes with the existing architecture for the BraTS 2021 datasets.**

| Author details | Methodology | DC | IoU | ACC |
|---|---|---|---|---|
| Alkassar, et al. [25] | CNN | 0.81 | 0.77 | 0.882 |
| Gadosey, Pius Kwao, et al [27] | SD-U-Net | 0.98 | 0.80 | 0.955 |
| Guvenc, et al. [31] | U-Net | 0.862 | 0.74 | 0.901 |
| Raza, Rehan, et al. [32] | Residual U-Net | 0.84 | 0.83 | 0.921 |
| Ahmad, Parvez, et al. [33] | Dense U-Net | 0.89 | 0.85 | 0.940 |
| **Proposed** | Permutate U-Net | **0.91** | **0.88** | **0.996** |

## 6. Conclusion

Segmentation of tumor regions always involves crucial processes, especially in brain tumor segmentation is more difficult by boundary region pixels in MRI and PET brain imaging having low sensitivity. To extract more important features with improved imaging quality is a prominent research area in accurate characterization in medical image analysis, where image fusion plays a indispensable role. To fuse MRI and PET images, the PCA algorithm is used, as it has been shown to yield better results than other fusion algorithms. It is also observed that the resultant fused have improved resolution, good contrast with detailed-edge information as given in Table 2. The resultant fused image is considered to evaluate the Permutate U-Net model in effective segmentation.

The model is designed from scratch to suit the fused input images, and its hyper parameters are selected after a series of experiments to produce the best results. The U-Net model comprises a network of five down and up sampling layers with a middle layer. As the number of layers increases, the number of filters allows the model to learn more complex relations. The increased number of layers also resulted in more skip connections, enabling the model to learn even complex relations and avoid overfitting. Finally, the model effectively segments the tumor region with than existing models in terms of dice-coefficient, accuracy, and IoU of 0.91, 0.88 and 0.996 respectively. Father to prove superiority of the model, the

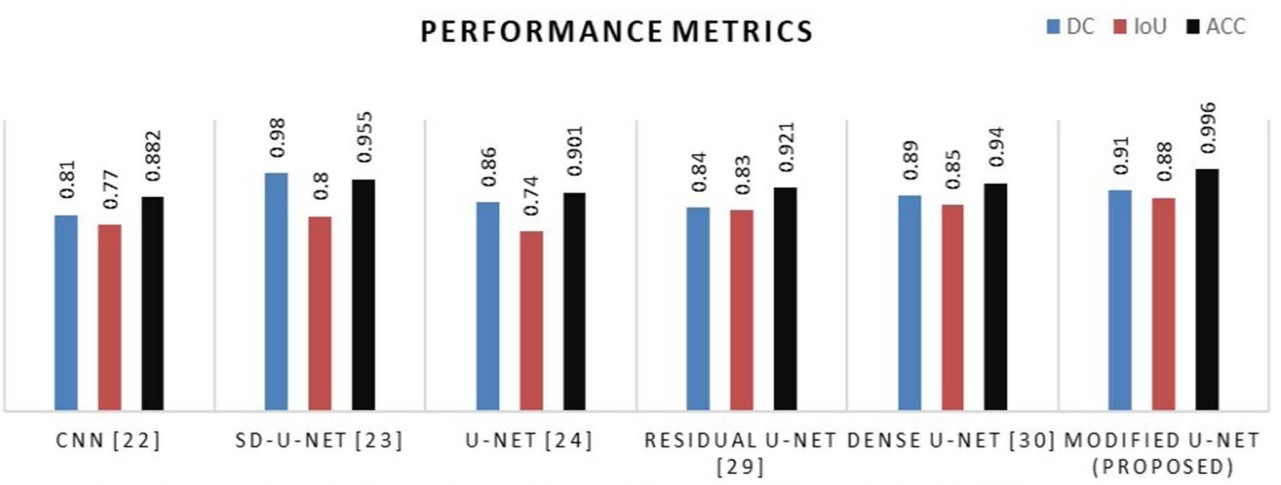

**Fig 12. The simulated results for 5-fold cross-validation.** (a) Dice Coefficient (b) IoU (c) Accuracy.

## PERFORMANCE METRICS   ■ DC  ■ IoU  ■ ACC

| | CNN [22] | SD-U-NET [23] | U-NET [24] | RESIDUAL U-NET [29] | DENSE U-NET [30] | MODIFIED U-NET (PROPOSED) |
|---|---|---|---|---|---|---|
| DC | 0.81 | 0.98 | 0.86 | 0.84 | 0.89 | 0.91 |
| IoU | 0.77 | 0.8 | 0.74 | 0.83 | 0.85 | 0.88 |
| ACC | 0.882 | 0.955 | 0.901 | 0.921 | 0.94 | 0.996 |

**Fig 13. Comparative analysis graph of proposed method with existing architecture for the BraTS 2015 datasets.**

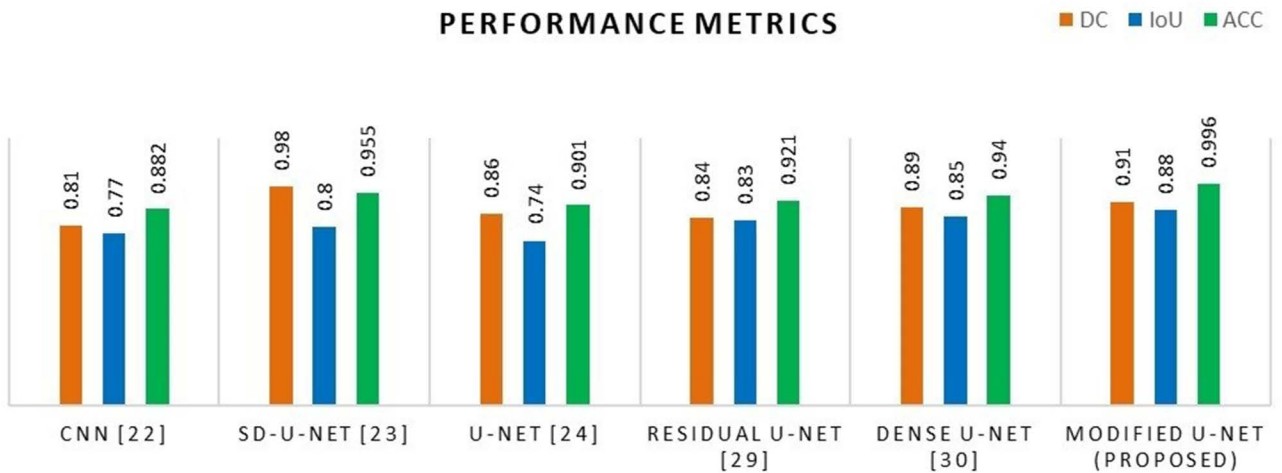

**Fig 14. Comparative analysis graph of proposed method with existing architecture for the BraTS 2020 datasets.**

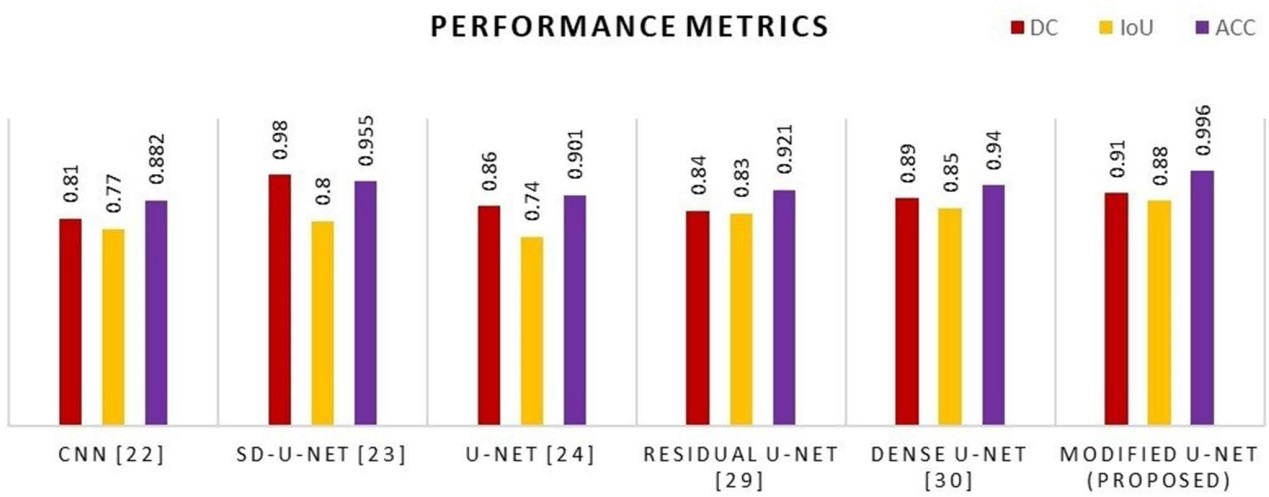

**Fig 15. Comparative analysis graph of proposed method with existing architecture for the BraTS 2021 datasets.**

resultant segmentation metrics of proposed model are compared by means of other existing DL architecture as shown in Tables 4–6 for various data sets.

Despite the modified U-Net architecture achieves promising results in brain tumour segmentation from fused MRI-PET images, certain limitations remain. Here the performance of model focusses mainly on quality of image fusion and the alignment between modalities which can introduce noise or artifacts affecting segmentation accuracy. Now a days many research focus on deep learning based fusion strategies for better image quality. So, there is scope for applying better deep learning for fusion strategies helps in improving the overall segmentation processes, which may enhance model robustness and clinical applicability.

## Author contributions

**Conceptualization:** Venu Allapakam, Peet Nalwaya.

**Data curation:** Saritha Saladi.

**Formal analysis:** Yepuganti Karuna, Venu Allapakam, Sk. Riyaz Hussian.

**Investigation:** S Priyanka.

**Methodology:** Venu Allapakam, Peet Nalwaya.

**Resources:** Yepuganti Karuna, Sk. Riyaz Hussian, Peet Nalwaya.

**Software:** S Priyanka.

**Supervision:** Yepuganti Karuna, Sk. Riyaz Hussian, Saritha Saladi.

**Validation:** S Priyanka, Saritha Saladi.

**Writing – original draft:** Venu Allapakam.

**Writing – review & editing:** Saritha Saladi.

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
