## [Decision Letter · Decision Letter 0]

13 Jun 2025

Dear Dr. Saladi,

Thank you for submitting your manuscript to PLOS ONE. After careful consideration, we feel that it has merit but does not fully meet PLOS ONE’s publication criteria as it currently stands. Therefore, we invite you to submit a revised version of the manuscript that addresses the points raised during the review process.

We look forward to receiving your revised manuscript.

Kind regards,

Ananth JP

Academic Editor

PLOS ONE

Additional Editor Comments (if provided):

Reviewers' comments:

Reviewer's Responses to Questions

**Comments to the Author**

1. Is the manuscript technically sound, and do the data support the conclusions?

Reviewer #1: Yes

Reviewer #2: Yes

2. Has the statistical analysis been performed appropriately and rigorously?

Reviewer #1: N/A

Reviewer #2: Yes

3. Have the authors made all data underlying the findings in their manuscript fully available?

Reviewer #1: Yes

Reviewer #2: Yes

4. Is the manuscript presented in an intelligible fashion and written in standard English?

Reviewer #1: No

Reviewer #2: No

Reviewer #1: The article on brain tumor segmentation using fused MRI-PET images and a modified U-Net framework is commendable. The integration of PCA for fusion and the comprehensive U-Net architecture is methodically laid out.

Here are some suggestions to strengthen your paper:

Clarify the term "Permutate U-Net" – What exactly is being permutated? Currently, the paper does not justify this terminology.

Please improve the language and grammar throughout the manuscript. Several sections are hard to follow due to sentence construction errors.

Include model complexity analysis – number of parameters, inference time, and training duration.

Consider using cross-validation or at least testing on an external validation set.

In comparative results, ensure that benchmarks are done on the same datasets. Add statistical tests to validate the improvements.

Visual fusion comparisons across methods (not just quantitative) would enhance understanding.

Some figures (especially block diagrams) could be redrawn for clarity and better resolution.

Reviewer #2: The title should be revised to include specific and detailed information reflecting the core contribution of the paper.

Keywords should be arranged in alphabetical order for clarity and consistency.

Avoid the use of first-person pronouns such as "our" and "we" to maintain an objective academic tone.

Ensure all sections are properly numbered. For example, "Contribution of the Proposed Work" should be listed as Section 2.

Maintain consistency and clarity in section headings throughout the paper.

The paper currently lacks a clear structure and reads more like lecture notes; a more formal academic organization is recommended.

The literature review section should incorporate recent research developments, as its current form lacks coverage of contemporary studies. Without a comprehensive and up-to-date survey, the novelty and relevance of the proposed work are significantly compromised.

**Do you want your identity to be public for this peer review?** For information about this choice, including consent withdrawal, please see our Privacy Policy

Reviewer #1: No

Reviewer #2: **Yes: ** Dr.Jasneet Kaur

---

## [Author Response · Author response to Decision Letter 1]

6 Oct 2025

We would like to thank all the reviewers for their valuable comments which helps us to improve our article both technically and language-wise.

---

## [Decision Letter · Decision Letter 1]

19 Oct 2025

Brain Tumour Segmentation in Fused MRI-PET Images With 

 Permutate U-Net Framework

PONE-D-25-25213R1

Dear Dr. Saladi,

We’re pleased to inform you that your manuscript has been judged scientifically suitable for publication and will be formally accepted for publication once it meets all outstanding technical requirements.

Kind regards,

Ananth JP

Academic Editor

PLOS ONE

Additional Editor Comments (optional):

Reviewers' comments:

Reviewer's Responses to Questions

**Comments to the Author**

Reviewer #1: All comments have been addressed

Reviewer #3: All comments have been addressed

2. Is the manuscript technically sound, and do the data support the conclusions?

Reviewer #1: Yes

Reviewer #3: Yes

3. Has the statistical analysis been performed appropriately and rigorously?

Reviewer #1: N/A

Reviewer #3: Yes

4. Have the authors made all data underlying the findings in their manuscript fully available?

Reviewer #1: Yes

Reviewer #3: Yes

5. Is the manuscript presented in an intelligible fashion and written in standard English?

Reviewer #1: Yes

Reviewer #3: Yes

Reviewer #1: The authors have clearly indicated that all datasets used in this study are publicly available and have provided appropriate links to open repositories

Reviewer #3: General Comments:

The authors present a well-designed and systematic approach to brain tumour segmentation using a novel Permutate U-Net architecture. The paper is technically sound, clearly written, and well-supported by quantitative evidence. The combination of PCA-based image fusion and modified U-Net structure demonstrates notable improvement over traditional U-Net and CNN models. The methodology, dataset description, and comparative results are thorough and well-documented.

Specific Comments for Minor Revision:

Please clarify how the permutation or modification of the U-Net differs structurally from Dense U-Net or Residual U-Net—this will help emphasize novelty.

Include a brief computational complexity or inference-time comparison between the proposed model and baseline U-Net.

Provide a short description of limitations or potential future improvements in the Conclusion section to strengthen the academic tone.

Ensure that all dataset citations are standardized with their DOIs or persistent links.

Proofread minor grammatical errors and unify capitalization in figure captions (e.g., “Figure 9 Segmented tumour region…”).

Overall Recommendation:

The manuscript is technically strong and contributes meaningfully to brain tumour segmentation research. It meets the PLOS ONE publication criteria, with only minor revisions recommended before acceptance.

**Do you want your identity to be public for this peer review?** For information about this choice, including consent withdrawal, please see our Privacy Policy

Reviewer #1: No

Reviewer #3: No

---

## [Editor Report · Acceptance letter]

PONE-D-25-25213R1

PLOS ONE

Dear Dr. Saladi,

I'm pleased to inform you that your manuscript has been deemed suitable for publication in PLOS ONE. Congratulations! Your manuscript is now being handed over to our production team.

Kind regards,

on behalf of

Dr. Ananth JP

Academic Editor

PLOS ONE